# Ambient-Pressured Acid-Catalysed Ethylene Glycol Organosolv Process: Liquefaction Structure–Activity Relationships from Model Cellulose–Lignin Mixtures to Lignocellulosic Wood Biomass

**DOI:** 10.3390/polym13121988

**Published:** 2021-06-17

**Authors:** Edita Jasiukaitytė-Grojzdek, Filipa A. Vicente, Miha Grilc, Blaž Likozar

**Affiliations:** Department of Catalysis and Chemical Reaction Engineering, National Institute of Chemistry, Hajdrihova 19, 1000 Ljubljana, Slovenia; miha.grilc@ki.si (M.G.); blaz.likozar@ki.si (B.L.)

**Keywords:** biomass, lignocellulose, lignin, cellulose, wood, liquefaction, organosolv fractionation, depolymerisation, ethylene glycol solvent, ^31^P NMR

## Abstract

Raising the awareness of carbon dioxide emissions, climate global warming and fossil fuel depletion has renewed the transition towards a circular economy approach, starting by addressing active bio-economic precepts that all portion amounts of wood are valorised as products. This is accomplished by minimizing residues formed (preferably no waste materials), maximizing reaction productivity yields, and optimising catalysed chemical by-products. Within framework structure determination, the present work aims at drawing a parallel between the characterisation of cellulose–lignin mixture (derived system model) liquefaction and real conversion process in the acidified ethylene glycol at moderate process conditions, i.e., 150 °C, ambient atmospheric pressure and potential bio-based solvent, for 4 h. Extended-processing liquid phase is characterized considering catalyst-transformed reactant species being produced, mainly recovered lignin-based polymer, by quantitative 31P, 13C and 1H nuclear magnetic resonance (NMR) spectroscopy, as well as the size exclusion- (SEC) or high performance liquid chromatography (HPLC) separation for higher or lower molecular weight compound compositions, respectively. Such mechanistic pathway analytics help to understand the steps in mild organosolv biopolymer fractionation, which is one of the key industrial barriers preventing a more widespread manufacturing of the biomass-derived (hydroxyl, carbonyl or carboxyl) aromatic monomers or oligomers for polycarbonates, polyesters, polyamides, polyurethanes and (epoxy) resins.

## 1. Introduction

Sustainability, industrial ecology, eco-efficiency and green chemistry are directing the development of the next generation of materials, products and processes. Biodegradable plastics and bio-based composites generated from renewable biomass feedstock are regarded as promising materials that could replace synthetic polymers and reduce global dependence on fossil resources [1,2]. Lignocellulosic (LC) biomass, wood in particular is among the most abundant renewable resources. Although wood has long been used as a raw material for construction and fuel, only over the last decades its potential for conversion into bio-fuels (e.g., bio-ethanol), production of commodity chemicals and biodegradable materials emerged [3,4,5]. As such, most research has been directed towards a more sustainable pathway to isolate cellulose, hemicellulose and lignin from biomass while finding green, cost-effective and efficient technologies to fractionate these compounds and apply them in several products [6,7,8].

Wood resources could be used to produce a wide range of valuable chemicals [9]. However, it is required to firstly depolymerize and convert the macromolecular components of wood into a liquid. This can be achieved through a thermochemical process via wood liquefaction, resulting in an effective transformation of LC biomass into high-added value intermediates like bio-based adhesives and polyols [10,11,12]. The use of liquefied wood in the preparation of new polymers is not only important for the efficient utilization of renewable resources, but also provides a way to replace the raw materials otherwise produced from crude oil.

One pathway to transform lignocellulosic biomass is its direct liquefaction, which is usually accomplished using a super- or subcritical fluid at high temperatures (250–400 °C) and high pressures (5–40 MPa) for a couple of hours [12]. A more cost-effective wood liquefaction could be achieved at atmospheric pressure however, in order to attain an appropriate degree of wood liquefaction, high acid concentrations are used to catalyse the process, which at the end affects the properties of the synthesized polymers [13]. An attractive alternative could be the recovery of the lignin-based material from the liquefied wood with the aim to use it as the “technical” lignin for instance for phenol-formaldehyde and/or epoxy resins or as bitumen substitute in asphalt converting primarily waste lignocellulosics into a feedstock [14,15].

Liquefaction bears a resemblance to an organosolv pretreatment as both are executed in acidified organic solvent. The key differences between those two pretreatments are the process duration, the acid concentration being used, the presence of water as a cosolvent and the final outcome. Organosolv pretreatment is designed to separate high-quality streams (cellulose, lignin and hemicellulose), while liquefaction or so-called “prolonged organosolv treatment” produces black liquor with negligible amounts of non-solubilized residue. Filtration coupled with precipitation produces a regular organosolv lignin and lignin-based polymer (LBP), respectively [16,17].

Due to the diversity in both chemical composition and reactivity of LC biomass components, the molecular structures in black liquor are quite complex, therefore, a comprehensive understanding of lignin-based polymer formation is essential to properly tailor the process for the further high value applications.

Herein, a parallel between “wood model” (physical mixture of cellulose-lignin (CL)) liquefaction and wood (W) liquefaction in acidified ethylene glycol (EG) was carried out in order to identify the reactions induced by cellulose and lignin as well as the ones emerged (or inhibited) due to the other wood constituents (hemicelluloses, tannins, waxes and resins) during the lignin-based polymer formation (CL_LBP_, W_LBP_). For this reason, beech sawdust and “wood model” composed of microcrystalline cellulose (MC) and beech milled-wood lignin (MWL) liquefactions were performed. MC was chosen to represent cellulose since its polymerization degree is similar to the one of the natural cellulose in wood, while the beech MWL was isolated from the same beech sawdust later used for the liquefaction. The ratio of cellulose and lignin in the “wood model” sample was maintained equivalent as in natural beech tree [18], and thus a sample composed of two parts of MC and one part of MWL, designated as CL, was examined in this study. Finally, by using the beech tree *Fagus Sylvatica* L., it was possible to valorise an abundant national resource.

## 2. Experimental Section

### 2.1. Materials

All chemicals used as analytical standards, namely 4-hydroxy-benzoic acid (≥99% purity), vanilic acid (≥97% purity), syringic acid (≥95% purity), coniferyl alcohol (98% purity), sinapyl alcohol (80% purity), vanillin (≥99% purity), syringaldehyde (98 % purity), furfural (99% purity), hydroxymethyl furfural (≥99% purity), glycerol guaiacol ether (≥98% purity), 3,5-dimethoxyphenol (99% purity), ferulic acid (≥99% purity) were purchased at Sigma Aldrich (St. Louis, MO, USA) as well as lithium bromide (99% purity), 1,3,5-trioxane (≥99% purity) and 2-chloro-4,4,5,5-tetramethyl-1,3,2-dioxaphospholane(95% purity). Beech tree (*Fagus Sylvatica* L.) sawdust was acquired from a central region of Slovenia. Milled-wood lignin (MWL) was isolated from beech tree (*Fagus Sylvatica* L.) according to the described procedures [19,20,21]. Microcrystalline cellulose (MC, white powder) with a molecular weight of 76,000 g mol^−1^, N-hydroxynaphthalimide (97% purity), chromium (III) acetylacetanoate (97% purity) were obtained from Acros Organics. Ethylene glycol (EG, 99.5% purity), sodium hydroxide (≥97% purity), methanol (99.9% purity), tetrahydrofurane (THF, ≥99% purity), pyridine (99.5% purity) and acetic acid (99.8% purity) were purchased from Merck (Darmstad, Germany). *p*-Toluene sulfonic acid monohydrate (PTSA, 98% purity), N,N-dimethylacetamide (DMA, 99.8% purity) and 2-chloro-1,3,2-dioxaphospholane (97% purity) were obtained from Fluka (Buchs, Swizerland).

### 2.2. Methods

#### 2.2.1. Cellulose-Lignin Mixture and Beech Sawdust (*Fagus sylvatica* L.) Liquefaction

Cellulose-lignin mixture (CL) composed from one part of beech MWL (1 g) and 2 parts of MC cellulose (2 g, dried at 105 °C for 24 h) and beech sawdust (24 mesh, 25 g, dried at 105 °C for 24 h; designated in this study as “wood” (W)) were liquefied with ethylene glycol in the ratio of 1:5, 15 g and 125 g, respectively. The liquefaction was carried out in a 100 mL or 500 mL rounded flask (two necks) equipped with a condenser. The reaction mixture was heated for 4 h at 150 °C in silicon oil bath under atmospheric pressure, while being constantly mixed with a magnetic or mechanical (four pitched blade turbine) stirrer (IKA Labortechnik, Staufen, Germany)at 400 rpm. To catalyse the reaction 3 wt% (based on EG) of PTSA was added to the reaction mixture. Samples from reaction mixture were taken at the different time intervals, immediately cooled in an ice-bath and neutralized with a 0.1 N NaOH solution to prevent further cellulose degradation prior to the characterization of the products.

After the removal of the non-liquefied residue (CL_R_, W_R_), the precipitation of soluble products (CL_SP_, W_SP_) in water enabled the separation of high-molecular weight material (CL_LBP_, W_LBP_) from low-molecular weight glycol fraction (CL_LGS_, W_LGS_). Thus, the prepared CL and W liquefied samples were studied by evaluating the changes in aromatic and aliphatic units as well as functional group content with time, by monitoring the change of molecular weights and through the identification of the low-molecular weight compounds.

#### 2.2.2. Sample Preparation

Samples for the analysis were prepared according to the procedure shown in Scheme 1. Each sample was diluted with excess amount of acetone, filtered using filter paper in order to isolate remaining, not liquefied residue. The filtered residue was repeatedly rinsed with 1,4-dioxane, acetone, water and pyridine until complete removal of the soluble products. Rinsed residues were dried at 60 °C under reduced pressure for 24 h.

#### 2.2.3. Measurement of Residual Cellulose-Lignin Mixture and Wood Content

The yield of the liquefaction was evaluated by determining the residual cellulose (CL_R_) content during cellulose-lignin mixture liquefaction and analogously residual wood (W_R_) content during the wood liquefaction. The residue content was determined as the weight of the obtained solids relative to the starting amount of cellulose-lignin mixture and starting amount of wood, respectively, as shown in Equation (1).
(1)$${\mathrm{CL}}_{R}\;\left(\%\right)=\frac{Z_{{\mathrm{CL}}_{R}}^{t}}{Z_{{\mathrm{CL}}_{R}}^{0}}\times 100$$
where $Z_{{\mathrm{CL}}_{R}}^{0}$ represents the initial weight of starting cellulose-lignin mixture and $Z_{{\mathrm{CL}}_{R}}^{t}$ the final weight of CL_R_. The same equation was used to determine the wood residue by replacing the CL mixture with wood.

#### 2.2.4. Determination of CL_LBP_ and W_LBP_ Content

The “lignin-based polymer” (LBP) content formed during the cellulose-lignin mixture liquefaction and during the wood liquefaction, expressed by percentage, was defined as Equation (2).
(2)$${\mathrm{CL}}_{\mathrm{LBP}}\;\left(\%\right)=\frac{Z_{{\mathrm{CL}}_{\mathrm{LBP}}}^{t}}{Z_{{\mathrm{CL}}_{\mathrm{SP}}}^{0}}\times 100$$
where $Z_{{\mathrm{CL}}_{\mathrm{SP}}}^{0}$ represents the initial weight of starting cellulose-lignin mixture in the soluble products (cf. Scheme 1) and $Z_{{\mathrm{CL}}_{\mathrm{LBP}}}^{t}$ the final weight of CL_LBP_. The same equation was used to determine the wood (W_LBP_) by replacing the CL mixture with wood.

#### 2.2.5. Nuclear Magnetic Resonance (NMR) Analysis

##### ^13^C NMR

Qualitative ^13^C NMR spectra of derivatives were recorded using a Unity Inova 300 Varian NMR spectrometer (Agilent Technologies, manufacturer name, Santa Clara, CA, USA) operating at 75 MHz. The measurements were conducted in DMSO-*d_6_* at 25 °C and tetramethylsilane (TMS) was used as an internal standard. Relaxation delay of 2 s was used between the scans. The free induction decay (FID) was multiplied by an exponential factor corresponding to 4 Hz line broadening prior to Fourier Transformation. For each spectrum, approximately 13,500 scans were accumulated. For quantitative ^13^C NMR, approximately 80 mg of acetylated (CL_LBP_, W_LBP_) sample and 0.8 mg of 1,3,5-trioxane used as internal standard were dissolved in 600 μL of deuterated chloroform (CDCl_3_-*d*). Acetylated samples were recorded with a Bruker 300 MHz spectrometer(Bruker, Billerica, MA, US) equipped with a Quad probe for ^31^P, ^13^C, ^19^F and ^1^H acquisition at 25 °C using an inverse gated decoupling pulse sequence. The chemical shifts were referred to the internal standard signal at 93.4 ppm. Relaxation delay of 12 s was used between the scans. Line broadening of 4 Hz was applied to FIDs before Fourier transformation of spectral data. For each spectrum, approximately 12,000 scans were accumulated.

##### Quantitative ^31^P NMR

Approximately 30 mg of LBP, CL_LBP_, W_LBP_ were transferred into sample vials, dissolved in 400 µL of pyridine and deuterated chloroform. (1.6:1 *v*/*v*). *N*–Hydroxynaphthalimide and chromium (III) acetylacetanoate were used as the internal standard and relaxation agent, respectively. 100 µL of the solution prepared from 0.1 mmol cm^–3^ of internal standard and 0.0143 mmol cm^–3^ of relaxation agent in the solvent system above was added to sample vials. Finally, 100 µL of phosphitylating reagent I (2-chloro-1,3,2,-dioxaphospholane) or reagent II (2-chloro-4,4,5,5-tetramethyl-1,3,2-dixaphospholane) was added and the mixture was left at room temperature for 1.5 h with continuous stirring. Then, the prepared sample solution was transferred into a 5 mm NMR tube. The spectra were recorded using a Bruker 300 MHz spectrometer equipped with a Quad probe for ^31^P, ^13^C, ^19^F and ^1^H acquisition. A sweep width of 10,000 Hz was observed, and spectra were accumulated with time delay of 25 s between pulses. A pulse width, causing 90° flip angle was used. Line broadening of 4 Hz was used in processing the spectra. All chemical shifts reported in this paper are relative to the reaction product of water with phosphitylating reagents (I or II) which have been observed to give sharp signals in pyridine/CDCl_3_ at 121.1 and 132.2 ppm, respectively [22,23].

The ^31^P NMR data reported in this work are averages of three experiments. The maximum standard deviation of our results was 2 × 10^−2^ mmol g^–1^, while the maximum standard error was 1 × 10^−2^ mmol g^–1^.

#### 2.2.6. Size-Exclusion Chromatography (SEC) Analysis

SEC of lignin-based polymer (LBP) and glycol samples (GS) was performed on a size-exclusion chromatographic system (HP–AGILENT system) equipped with a UV detector (Agilent Technologies, Santa Clara, CA, USA) set at 280 nm. Analyses were carried out at ambient temperature using THF and DMA with 0.01 M LiBr as eluents at a flow rate of 1 mL min^–1^. Aliquots (100 μL) of each lignin-based polymer sample (CL_LBP_, W_LBP_), dissolved in THF (1.5 mg cm^–3^), and glycol sample (CL_GS_, W_GS_), dissolved in DMA with 0.01 M LiBr, were injected into PLgel 3 µm MIXED E 7.5 × 300 mm^2^ and Polar Gel L 8 µm 7.5 × 300 mm^2^, respectively. The columns specifications allow separation of molecular weights up to 3.0 × 10^4^ g mol^–1^. The SEC system was calibrated with polystyrene standards in the molecular weight range of 500 g mol^–1^ to 3.0 × 10^4^ g mol^–1^ for CL_LBP_, W_LBP_ samples and in the range of 100 g mol^–1^ to 3.0 × 10^4^ g mol^–1^ for CL_GS_, W_GS_ samples. The chromatographic data was processed with PSS (Polymer Standards Service) WinGPC Unity software (PSS, Amherst, MA, USA).

#### 2.2.7. High-Performance Liquid Chromatography (HPLC) Analysis

The detection of lignin phenols was carried out on a PerkinElmer HPLC system (PerkinElmer, Waltham, MA, USA) consisting of a binary pump (Series 200 LC) and a diode array detector (L-235) set at 280 nm. Methanol-water (25:75, *v*/*v*) with 1% (*v*/*v*) addition of acetic acid was used as a mobile phase at a flow rate of 1 cm^3^/min at ambient temperature. Aliquots (10 μL) of each glycol sample (LGS, CL_GS_, W_GS_) were injected into a stainless-steel column (200 × 4.6 mm^2^) packed with ODS Hypersil (5 µm). The chromatographic data were processed with PSS (Polymer Standards Service) WinGPC Unity software [24].

## 3. Results

Unlike most works, we focused this research on wood liquefaction using milder conditions—namely, 150 °C, and atmospheric pressure but for longer processing time (4 h) to achieve an efficient wood liquefaction and lignin-based polymer recovery. The main aim of this study was to characterize this extended liquefaction considering all the products being formed. Hence, multiple techniques were applied, for instance NMR spectroscopy, in particular quantitative ^31^P and ^13^C NMR, and SEC and HPLC for higher and lower molecular weight compounds, respectively (Figure 1).

### 3.1. CL and W Liquefaction–Characterization of Remained Residue and Obtained LBP

During the CL and W liquefaction, samples were taken in different intervals in order to follow the amount of residue that remained and the amount of LBP that was obtained, as shown in Figure 2. From this figure, it is evident a gradual decrease in the CL and W residue that remains after different periods of liquefaction, indicating the slow degradation of cellulose micro-fibrils followed by the synchronous dissolution of the accessible lignin. The initial decrease of residual wood content within the first 15 min of liquefaction implies the conversion of approximately 60% of starting wood into soluble products. In contrast, in the case of CL liquefaction, only 25% of starting CL mixture was solubilized. This could be attributed to the differences in the crystallinity of MC and natural cellulose in wood as natural cellulose presents lower crystallinity (CrI ~45%) [25] and is more susceptible to degradation in acid-catalysed EG than the highly crystalline MC (CrI ~85%) [26]. Similar profiles of the solid residues with liquefaction time were obtained after the ionic liquid-catalyzed wood solvolysis in glycerol where the initial wood liquefaction was referred to the rapid hemicellulose dissolution [27]. Obviously, beside the aforementioned cellulose susceptibility/crystallinity differences, the fast hemicellulose solubilization further affected the prompt drop of the residual solids. Nevertheless, at approximately 90 min of reaction, the liquefaction of both CL and W started to behave in a similar manner, showing a linear and smooth decrease until the end of the liquefaction, where ~10% of residue remained. Taking into account an absolute MWL solubility in acidified EG as well as MC liquefaction of up to 99% during the 4 h treatment [28], the obtained CL_R_ and W_R_ residues imply the presence of unknown factors encumbering the complete CL and W conversion into soluble products. The formation of insoluble residue has been reported to be a result of induced mutual reactions between depolymerized cellulose and aromatic lignin derivatives [29].

Beside the gradual degradation of cellulose micro-fibrils followed by the synchronous dissolution of the accessible lignin during the wood liquefaction, the increasing amounts of recovered CL_LBP_ and W_LBP_ with time (Figure 2II) imply the presence of reactions between the solubilized products. The content of the recovered CL_LBP_ and W_LBP_ increased by ~25% by the end of the liquefaction. This could be the result of both a more extensive EG chain as well as the presumable cellulose and wood degradation products incorporation into the lignin backbone.

### 3.2. Quantitative NMR Analysis of CL and W Liquefaction

The initial assumption regarding lignin’s modification was confirmed by ^31^P and ^13^C NMR. The structural changes of precipitated lignin and possible incorporation of EG into lignin backbone were examined by quantitative ^31^P and ^13^C NMR spectroscopy. It should be noted that ^13^C NMR did not show any signals relative to the cellulose and hemicellulose, thus there was no formation of lignin-carbohydrate complexes that could be determined in the precipitated liquid wood sample [30,31]. The typical lignin spectra obtained by ^31^P and ^13^C NMR analyses (Figure 3 and Figure 4) indicate a complete lignin isolation from liquefied carbohydrates. Therefore, the precipitated liquid wood sample (W_LBP_) as well as the precipitated liquefied cellulose-lignin sample (CL_LBP_) were referred as the “*lignin-based polymer*”. Figure 3 displays the ^31^P NMR spectra, while Table 1 contains the quantitative analysis of functional groups present in the samples with the liquefaction time. In both cases, MWL data was added as a reference.

The spectra of samples derivatized with 2-chloro-4,4′,5,5′-tetramethyl-1,3,2-dioxaphospholane showed a reduction of overall aliphatic OH groups from 7.44 mmol g^−1^ to 3.15 mmol g^−1^ and 2.72 mmol g^−1^ in CL_LBP_ and W_LBP_, respectively. The sharp decrease of overall aliphatic OH content at the beginning of the treatment as well as in the case of MWL liquefaction indicates the initial lignin degradation followed by the formation of soluble low-molecular weight derivatives. Furthermore, the lower aliphatic OH content observed after 15 min of treatment in W_LBP_ may be attributed to the more extensive side degradation of wood lignin than in the case of isolated MWL and MC mixture liquefaction.

From the spectra of samples derivatized with 2-chloro-1,3,2-dioxaphospholane (Figure 2), the substitution of *α*-hydroxyls in lignin macromolecule is confirmed by the complete elimination of broad signals in the range of 136.2–134.5 ppm. The quantification of relative primary (P–OH) and secondary (S–OH) hydroxyl content enabled the determination of *S–OH/P–OH* ratio decrease with treatment time (Table 1). This observation may imply the introduction of EG molecule at Cα position, similarly as was revealed in the case of MWL liquefaction [24]. By comparing the observed *S–OH/P–OH* values, it is evident that more pronounced *α*-hydroxyl substitution was determined during MWL liquefaction (0.09) followed by W liquefaction (0.12) and CL (0.16). Furthermore, the increase of *S–OH/P–OH* values in the samples with cellulose (CL_LBP_ and W_LBP_) may imply the substitution of P–OH by 2-hydroxyethyl levulinate obtained after cellulose liquefaction. Accordingly, the reduced P–OH content can result in an increase of *S–OH/P–OH* ratio at the end of CL and W liquefaction.

The distribution and quantitative evaluation of phenolic OH and carboxylic acids were also obtained by ^31^P NMR for the samples derivatized with 2-chloro-4,4′,5,5′-tetramethyl-1,3,2-dioxaphospholane (Figure 2). The CL and W liquefaction caused an increase of overall phenolic OH content only during the initial 15 min of treatment that is from 1.11 mmol g^–1^ to 4.87 mmol g^–1^ and 4.49 mmol g^–1^ for CL_LBP_ and W_LBP_, respectively. That may be attributed to the cleavage of β-ethers in lignin macromolecule. Phenolic OH content tends to decrease during the prolonged treatment to 3.74 mmol g^–1^ and 3.14 mmol g^–1^, respectively, indicating the occurrence of additional lignin condensation reactions with 2-hydroxyethyl levulinate, obtained after cellulose liquefaction. During the initial 15 min of reaction, the more pronounced increase of the syringyl (Sr) phenolic units (from 0.29 mmol g^–1^ to 2.66 mmol g^–1^ in CL_LBP_-15 and to 2.95 mmol g^–1^ in W_LBP_-15) than in the guaiacyl and *p*-hydroxyphenyl (G + H) phenolic units (from 0.89 mmol g^–1^ to 2.21 mmol g^–1^ in CL_LBP_-15 and to 1.54 mmol g^–1^ in W_LBP_-15) indicates a more extensive cleavage of β-aryl ethers by syringyl moieties than those by guaiacyl moieties. Moreover, the continuous decrease of total OH content during CL and W liquefaction could be the result of both the occurrence of additional condensation reactions induced by cellulose liquefaction products and possible presence of lignin self-polymerization in the acidic medium [32].

The aliphatic/aromatic OH ratio of CL_LBP_ and W_LBP_ samples significantly decreased during the treatment, suggesting a preferential degradation of aliphatic side chains in comparison with aromatic units. The decrease of aliphatic OH/aromatic OH ratio from 6.70 to 0.84 (CL) and 0.87 (W) indicated a radical change of the sample nature that reflects in predominance of aromatic OH. Comparing the determined CL and W aliphatic OH/aromatic OH ratios with the one of MWL (0.49), it is evident that CL_LBP_ and W_LBP_ samples possessed a lower content of aromatic OH due to presumable condensation reactions between cellulose liquefaction products and lignin.

The carboxylic acid groups showed a clear decrease during both the CL and W liquefactions that accordingly implies the presence of esterification reactions induced by EG. However, the appearance of the detectable amounts of COOH after 120 min of treatment may be attributed to the hydrolysis of ester linkages by water molecules obtained as a side product after etherification of OH groups and/or the possible incorporation of syringic and vanillic acids obtained after the initial lignin degradation.

Regarding the ^13^C NMR quantitative analysis, the spectra are shown in Figure 3 while the respective structural analysis is presented in Table 2. The chemical structures of all steps, the proposed reaction mechanism and end products described in this table are also represented in Appendix A. It should be stressed that an inverse gated decoupling sequence was used for the quantitative estimation of lignin structures (CL_LBP_ and W_LBP_) formed during the liquefaction, particularly the incorporation of aliphatic chains into the lignin backbone. Here, the spectra of CL_LBP_ and W_LBP_ samples show only Cγ signal in the aliphatic carbon region indicating a preferential cleavage of β-aryl ether bonds in lignin. Furthermore, due to the similar overlapping issue of Cγ signal at 62.6 ppm with the other signals that may correspond to both the etherified EG and cellulose liquefaction products, the overall aliphatic carbon amount in CL_LBP_ and W_LBP_ samples was evaluated by the integration of all signals in the range of 59.5–69.6 ppm.

The increase of the aliphatic carbon content with liquefaction time from 3.0 mmol g^–1^ to 3.3 mmol g^–1^ (CL_LBP_) and 3.7 mmol g^–1^ (W_LBP_) indicates aliphatic chain introduction into the lignin structure (Table 2). The less pronounced aliphatic chain incorporation into the lignin backbone during the CL liquefaction may be explained by the partial EG involvement in highly crystalline cellulose degradation where the formation of the soluble EG-glucoside and formation of levulinic acid ester proceeded relatively more slowly than the degradation of natural cellulose in wood. Accordingly, the more rapid degradation of the natural cellulose produced levunilic acid ester at an early reaction stage and was possibly involved in the formation of the final W_LBP_ sample. Thus, the higher content of aliphatic carbons determined in W_LBP_ sample was presumably the result of the EG chain and levunilic acid ester incorporation into the lignin macromolecule. In addition, the presence of levulinic acid ester in W_LBP_ sample structure was confirmed by the appearance of the well resolved signal at 66.9 ppm that corresponds to the C6 carbon in 2-hydroxyethyl levulinate [28] (Figure 4).

The amount of aromatic units was determined by integrating the signals in the range from 101.0 to 154.8 ppm and increased from 3.4 mmol g^–1^ to 4.1 mmol g^–1^ (CL_LBP_) and to 4.0 mmol g^–1^ (W_LBP_). The obtained results imply that during the liquefaction of samples with cellulose, the reduced lignin functionalization by EG and by levulinic acid ester affects CL_LBP_ and W_LBP_ composition by similarly increasing the relative amount of aromatic units. In contrast to the CL_LBP_ and W_LBP_ samples, LBP contained a reduced amount of aromatic units at the end of the treatment indicating a more pronounced incorporation of aliphatic moieties into the lignin structure.

The methoxy group content evaluated by the integration of the signal at 55.6 ppm showed a decrease from 5.3 mmol g^–1^ to 1.6 mmol g^–1^ (CL_LBP_) and 1.8 mmol g^–1^ (W_LBP_) at the end of the liquefaction. The decrease in the intensity of the signal corresponding to the methoxy groups in CL_LBP_ and W_LBP_ followed by synchronous aromatic unit increase implies the occurrence of demethoxylation or demethylation reactions of guaiacyl and syringyl units within the lignin macromolecule.

### 3.3. SEC Analysis

SEC analysis was performed to evaluate the changes in molecular weight distribution (MWD) of the isolated CL_LBP_, W_LBP_ and lignin-glycol samples (CL_LGS_, W_LGS_). Samples for SEC analysis were prepared by separating low-molecular weight lignin derivatives and cellulose liquefaction products according to the sample preparation scheme (Scheme 1) being the results displayed in Figure 5 and Table 3 and Table 4.

The acetylated MWL sample with average molecular weight ($\bar{M_{w}}$) of 5800 g mol^–1^ was used as a reference material for the obtained CL_LBP_ and W_LBP_ samples. The decrease of $\bar{M_{w}}$ observed during the first 5 min of the CL liquefaction indicates the initial MWL degradation (Table 3). The increased aromatic region absorbance at 280 nm and the shift of the main peak towards the lower molecular weight of CL_LBP_-5 chromatogram shown in Figure 5I) confirms the formation of lower molecular weight CL_LBP_ fragments. In addition, due to the slow MC liquefaction, CL_LBP_ samples at the beginning of the treatment refer only to the change of MWL molecular weight. In contrast to the CL_LBP_-5, the increase of $\bar{M_{n}}$ from 1020 g mol^–1^ to 1400 g mol^–1^ and intensively reduced aromatic region absorbance at 280 nm of the CL_LBP_-15 chromatogram, indicates the formation of larger structures followed by the aliphatic chain incorporation into the lignin structure. Furthermore, the decrease of the overall aliphatic OH content followed by *S-OH/P-OH* ratio decrease, as determined by quantitative ^31^P NMR, implies the condensation reactions between highly reactive *α*-aliphatic OH in the lignin macromolecule and an excess of aliphatic OH provided by EG, like in the MWL liquefaction [24]. Additionally, the reduction in the CL_LBP_-15 main peak intensity could be attributed to the synchronous lignin degradation and the formation of low-molecular weight fragments that consequently increased the absorbance in the aromatic region at 280 nm.

After 15 min of the liquefaction, the continuous growth of $\bar{M_{w}}$ and the decrease of overall aliphatic OH content determined by quantitative ^31^P NMR imply the presence of condensation reactions involving less reactive primary aliphatic OH groups at Cγ position in lignin phenyl propane units and/or those obtained after EG incorporation. The formation of the larger CL_LBP_-120 fragments with $\bar{M_{w}}$ of 3800 g mol^–1^ followed by enlarged PDI from 1.9 to 8.1 and $\bar{M_{n}}$ reduction from 1400 g mol^–1^ to 470 g mol^–1^ indicates, beside the large CL_LBP_-120 structures, the occurrence of lower molecular weight fragments. The increased absorbance at 280 nm in the CL_LBP_-120 chromatogram (Figure 5I) confirms the presence of fragments formed by condensation reactions between EG, non-etherified syringyl and guaiacyl units and levulinic acid.

After the total 240 min of the CL liquefaction the increased $\bar{M_{w}}$ of CL_LBP_-240 to 5050 g mol^–1^ and the simultaneously reduced $\bar{M_{n}}$ to 340 g mol^–1^ implies the occurrence of concurrent degradation and condensation processes. The increase of the aromatic unit content, the invariability of aliphatic carbon content and the gradual decrease of phenolic OH content after 120 min of treatment, determined by quantitative ^13^C and ^31^P NMR, implies that CL_LBP_-240 high-molecular weight fraction is formed by inter-condensation of the present CL_LBP_ fragments via phenolic OH without additional EG chain incorporation. The broad CL_LBP_-240 MWD and the reduced absorbance at 280 nm evident in the chromatogram of Figure 5I) indicate the occurrence of lower-molecular weight CL_LBP_ fragments that could be the result of the simultaneous degradation of the newly formed CL_LBP_ macromolecules.

The trend for low-molecular weight lignin degradation products to condense with EG within CL_LGS_ can be evaluated from the gradual PDI decrease from 8.6 to 7.6 and the appearance of a high-molecular weight shoulder during 120 min of the treatment (Table 3; Figure 5II). Moreover, the formation of larger CL_LGS_ fragments and the simultaneous conversion into the CL_LBP_ constituents cannot be deduced from the obtained SEC data due to the appearance of low-molecular weight cellulose liquefaction products after 120 min of the treatment that extensively enlarged the PDI value (to 12.9) at the end of the reaction.

Molecular weights and polydispersity indices of W_LBP_ and W_LGS_ samples determined by SEC are summarized in Table 4. The $\bar{M_{w}}$ of the W_LBP_-15 sample was determined to be 2650 g mol^–1^, which is similar to the one of the CL_LBP_-15 sample (2600 g mol^–1^), hence indicating the synchronous degradation of solubilized lignin within the initial 15 min of the wood liquefaction, in addition to the more pronounced natural cellulose liquefaction. Furthermore, the induced aromatic region absorbance at 280 nm and the shift of the main peak towards the lower molecular weight of the W_LBP_-15 chromatogram shown in Figure 5II) confirms the formation of W_LBP_-15 smaller fragments. In contrast to the $\bar{M_{w}}$ resemblance between CL_LBP_-15 and W_LBP_-15, the reduced value of $\bar{M_{n}}$ (820 g mol^–1^; W_LBP_-15) may be interpreted in terms of the more pronounced in situ lignin susceptibility to the degradation during the treatment.

The continuous $\bar{M_{w}}$ growth and the formation of the larger W_LBP_-120 structures proceeded analogously as during the CL liquefaction. Thus, the increase of the $\bar{M_{w}}$ to 7400 g mol^–1^, the PDI enlargement from 3.3 to 15.4 and the $\bar{M_{n}}$ decrease from 820 g mol^–1^ to 480 g mol^–1^ indicated the formation of a highly polydispersed product (Table 4). The more pronounced PDI enlargement and the $\bar{M_{w}}$ increase compared to those obtained after the CL liquefaction (PDI—from 1.9 to 8.1 and $\bar{M_{w}}$—from 2600 g mol^–1^ to 3800 g mol^–1^). The introduction of additional aliphatic structures into W_LBP_-120 was also confirmed by the extensive reduction of absorbance at 280 nm in the W_LBP_-120 chromatogram (Figure 5III).

At the end of W liquefaction, $M_{w}$ of the W_LBP_-240 sample increased to 14,650 g mol^–1^, resembling the CL treatment. The decrease of the aliphatic carbon content, the invariability of the aromatic unit content and the gradual decrease of the overall phenolic OH content after 120 min of reaction (determined by quantitative ^13^C and ^31^P NMR analysis) show that the high-molecular weight of the W_LBP_-240 fraction is formed by inter-condensation of the present W_LBP_ fragments via phenolic OH without additional EG incorporation. However, in contrast to the CL liquefaction, the formation of extremely polydispersed W_LBP_-240 with a PDI of 38.8 and a $M_{n}$ of 380 g mol^–1^ may be attributed to the occurrence of structures formed after the condensation between EG, low-molecular weight lignin degradation products and the compounds obtained after cellulose liquefaction in W_LGS_. The increased absorbance in the aromatic region (280 nm) of the W_LBP_-240 chromatogram (Figure 5III) confirms the presence of fragments formed by condensation reactions between EG, non-etherified syringyl and guaiacyl units and levulinic acid. Nevertheless, the formation of such polydispersed product at the end of the W liquefaction might be obtained due to the presence of the natural wood components such as hemicellulose, resins and waxes.

The $M_{w}$ and PDI of W_LGS_ samples with liquefaction time are listed in Table 4. The $M_{n}$ decrease from 65 g mol^–1^ to 35 g mol^–1^ followed by a PDI increase from 9.6 to 16.9 during the initial 60 min of the reaction points towards formation of a number of low-molecular weight compounds. Besides the lignin degradation products such as non-etherified syringyl and guaiacyl units, the W_LGS_ is enriched by the cellulose degradation products and solubilized and partially degraded hemicellulose. The occurrence of the synchronous condensation reactions within W_LGS_ during 120 min of the liquefaction can be deduced from the slightly reduced PDI value (W_LGS_-120—Table 4). Furthermore, the gradual $M_{w}$ decrease with reaction time show that the larger W_LGS_ fragments with increased $M_{w}$ were converted into W_LBP_ constituents and simultaneously eliminated from W_LGS_, resulting W_LBP_-120 with the enlarged PDI. The increased intensity of the peak at the low-molecular weight in W_LGS_-240 chromatogram shown in Figure 5IV) may indicate the formation of the levulinic acid ester and lignin degradation products. Accordingly, due to the increased PDI value (to 10.4) at the end of the treatment, the formation of larger W_LGS_ fragments cannot be determined similarly, as in the case of the CL liquefaction.

### 3.4. HPLC analysis of the Low-Molecular Weight Products

The isolated lignin-glycol samples obtained after the CL and W liquefaction (CL_LGS_ and W_LGS_) were subjected to HPLC analysis in order to identify the low-molecular weight cellulose and lignin derived products. HPLC chromatograms of CL_LGS_-5 and CL_LGS_-240 shown in Figure 6II,III were obtained under the same experimental conditions as applied for LGS samples. The use of standard compounds, whose structures are displayed in Figure 6I, confirmed the presence of furfural (2), 4-hydroxy-benzoic acid (3), vanillic acid (4), syringic acid (5), vanillin (6), coniferyl and sinapyl alcohols (7, 9), syringaldehyde (8) and ferulic acid (13) in CL_LGS_-5, repeatedly confirming a fast initial lignin and cellulose degradation. Due to the prompt reactions, not all peaks have been identified. Nonetheless, this data still allowed for iteractive conclusions.

From the CL_LGS_-240 chromatogram shown in Figure 6III, it is evident that the monomeric lignin-derived compounds were not extensively modified. However, the change of the peak intensities confirms them being reactive with liquefaction intermediates. In addition, the appearance of the peaks at the longer retention times indicates the formation of the larger structures obtained after low-molecular weight product condensation with EG or between each other, while the intensive peaks eluted at shorter retention time may be attributed mainly to the MC liquefaction products.

HPLC chromatograms of W_LGS_-15 and W_LGS_-240 are presented in Figure 6IV,V). The use of standard compounds enabled identification of lignin degradation products such as 4-hydroxy-benzoic acid (3), vanillic acid (4), syringic acid (5), vanillin (6), coniferyl and sinapyl alcohols (7, 9), syringaldehyde (8), glycerol guaiacol ester (11) and ferulic acid (13) in W_LGS_ taken after 15 min of the liquefaction. The identified lignin degradation products found in traces in W_LGS_-15 imply that the lignin isolated during the reaction from wood sawdust undergoes a similar degradation as MWL. The reduced amounts of the lignin degradation products in W_LGS_-15 indicates the occurrence of a more pronounced condensation of low-molecular weight components presumably due to their pronounced reactivity. Accordingly, the formation of the W_LBP_-15 fragments with reduced $M_{n}$ (820 g mol^–1^) compared to the ones of CL_LBP_-15 (1400 g mol^–1^) may be interpreted in terms of a rapid lignin-derived component reaction with the reaction intermediates forming a number of smaller though EG-insoluble W_LBP_-15.

Compared to CL_LGS_-15, the negligible amount of vanillin (6) detected in W_LGS_-15, the presence of glycerol guaiacol ether (11) and vanillic acid (4) may imply presumable vanillin consumption in order to produce the mentioned compounds (4 and 11). Moreover, the reduced intensity of the peak corresponding to the coniferyl alcohol (7) and the absence of sinapyl alcohol (9) might indicate their participation in W_LBP_-15 formation followed by extensive monolignol (7 and 9) degradation.

The identification of syringic acid (5) and the absence of syringaldehyde (8) in the W_LGS_-240 chromatogram (Figure 6V) could be interpreted in terms of syringaldehyde oxidation. Consequently, and excepting syringic acid, other peaks in the W_LGS_-240 chromatogram were not identified with the standards as compounds, recognized at the beginning of the reaction were modified after the reaction with EG and reaction intermediates.

In contrast to the W_LGS_-240, peaks observed at longer retention times in CL_LGS_-240 imply the presence of EG-soluble lignin-derived dimers and trimers, while the intensive peaks at shorter retention times in both CL_LGS_-240 and W_LGS_-240 chromatograms originate from cellulose liquefaction products, in W_LGS_-240 additionally from degraded hemicelluloses.

### 3.5. Quantitative NMR Analysis of the Low-Molecular Weight Products

The qualitative ^13^C NMR analyses of CL_LGS_ and W_LGS_ were performed in order to confirm the presence of lignin degradation products previously detected by HPLC analysis, being the results depicted in Appendix A and Table 5 and Table 6.

Qualitative ^13^C NMR spectra of CL_LGS_-5 and CL_LGS_-240 are depicted in Appendix A. The intensive signals observed in the aliphatic carbon region correspond to the aliphatic structures in MC and MWL degradation products, compounds obtained due to the latter’s interactions with EG, EG-derived products and free EG. Therefore, the presence of MC and MWL degradation products in CL_LGS_ detected by HPLC was confirmed by the existence of the corresponding signals in the range from 104.0 to 210.0 ppm. The signals in the determined range were designated according to literature [18,30,33] and are summarized in Table 5.

The characteristic signals of etherified and non-etherified (phenolic free) guaiacyl and syringyl moieties in CL_LGS_-5 ^13^C NMR spectrum confirm the presence of compounds identified by HPLC. The more pronounced MWL degradation during the CL liquefaction could also be deduced from the observed intensive signals 6, 8, 9 corresponding to coniferyl and sinapyl alcohol, non-etherified syringyl and etherified guaiacyl subunits. Furthermore, the increased intensity of 6, 8, 9 signals in the CL_LGS_-240 spectrum at the end of the reaction implies a continous appearance of identified substructures with time. Esterification and etherification of the free phenolic guaiacyl and syringyl subunits is indicated by the absence of signals 10, 7, 16 and the newly appeared signal 12, corresponding to the carbonyl groups in aromatic esters in the CL_LGS_-240 spectrum (Appendix A). The presence of 17 and 18 signals in the CL_LGS_-240 spectrum implies the occurrence of coniferyl and sinapyl alcohol oxidation to aldehyde and the oxidation of *α*-hydroxyl group in the phenyl propane units, respectively. Besides the lignin-derived structures, MC liquefaction products are indicated by the appearance of intensive 14, 15 and 19 signals corresponding to the carbonyl groups in hydroxyethyl levulinate and aliphatic carboxylic acid. Moreover, the intensive peaks observed in the CL_LGS_-240 HPLC chromatogram (Figure 5III) could be related to the hydroxyethyl levulinate, which just like the aromatic lignin-derived products absorbs UV light at 280 nm.

Qualitative ^13^C NMR spectra of W_LGS_-15 and W_LGS_-240 are depicted in Appendix A. The presence of the analogous intensive signals in the aliphatic carbon region as in CL_LGS_ spectra indicates the presence of aliphatic structures in cellulose and lignin degradation products, compounds obtained due to the latter interactions with EG, EG-derived products and free EG. Therefore, signals in the range from 95.0 to 210.0 ppm were used to confirm cellulose and lignin degradation products in W_LGS_. The signals in the determined range are summarized in Table 6 [18,30,33].

The more pronounced cellulose degradation during the initial 15 min of wood liquefaction than during the CL treatment could be deduced from the presence of the signals 2, 3 and 16 that correspond to EG α-, β-D-glucopyranosides and to carbonyl groups in furfural-based structures, respectively. The decreased intensity of signal 16 in the W_LGS_-240 spectrum indicates the modification of furfural-based structures by reaction intermediates, while the appearance of signals 15 and 18 corresponding to the carbonyl groups in hydroxyethyl levulinate confirms the degradation of EG-glucosides with the reaction time. Similarly, the formation of levulinate was confirmed with FTIR analysis during the one-pot solvolysis and hydro-treatment of lignocellulosic biomass reported by Grilc et al. [34].

Lignin degradation products in W_LGS_-15 are confirmed by the presence of signals (4, 6, 9, 10, 12) corresponding to non-etherified and etherified guaiacyl and syringyl units. The occurrence of coniferyl alcohol and ferulic acid can be deduced from the existence of signals 8 and 14. Moreover, the absence of signal 8 and the accordingly increased intensity of signal 13 in W_LGS_-240 spectrum may be interpreted in terms of coniferyl alcohol degradation followed by the formation of vanillic acid and vanillic acid ester. The etherification of free phenolic guaiacyl and syringyl subunits with time is confirmed by *i)* the absence of signal 10 and the reduced intensity of signal 12, and *ii)* the newly appeared signal 11 and the increased signal 9 intensity, both corresponding to the etherified syringyl subunits in W_LGS_-240. The increase of signal 17 intensity with the treatment time confirms the gradual oxidation of *α*-hydroxyl group in phenyl propane units, while the extensive reduction of signal 16 in W_LGS_-240 may be attributed to the oxidation of furfural-based structures.

From the assignation of well resolved signals in W_LGS_ spectra, it can be assumed that the lignin isolated during the W liquefaction contains more reactive functional groups than the MWL. Therefore, this explains the detection of more lignin-derived products (syringic acid, syringaldehyde, vanillin, vanillic acid, coniferyl alcohol and ferulic acid) in CL_LGS_-240 than in W_LGS_-240 (syringic acid). HPLC confirms the higher reactivity of the low-molecular weight lignin-derived structures obtained during the W liquefaction as well. Furthermore, due to the presence of the analogous well-defined signals (15, 18) in the W_LGS_-240 spectrum as signals (14, 19) in CL_LGS_-240 spectrum, the intensive peaks observed in W_LGS_-240 HPLC chromatogram (Figure 5V) can be related to the hydroxyethyl levulinate, which, like aromatic lignin-derived products, tends to absorb UV light at 280 nm.

## 4. Conclusions

This study aimed at understanding the formation of ethylene glycol organosolv lignin (lignin-based polymer) isolated via a prolonged organosolv treatment (liquefaction). This was achieved by an overall characterization of the obtained products during an extended cellulose-lignin mixture (wood model) liquefaction and a real wood liquefaction in acidified ethylene glycol at moderate conditions, i.e., 150 °C and ambient atmospheric pressure, for 4 h. By detecting and quantifying the change of the aliphatic, phenolic and carboxylic OH group content during the liquefaction by quantitative ^31^P NMR, it was possible to monitor the simultaneous lignin extraction and degradation followed by the formation of soluble low-molecular weight derivatives that were further characterized through HPLC. Moreover, the quantitative ^13^C NMR analysis revealed the gradual introduction of EG moieties into the lignin structure, affecting its overall composition by lowering the relative amount of aromatic units. Moreover, the aliphatic part of the lignin-based polymer was additionally enriched with the moieties of the levunilic acid, the main cellulose liquefaction product. At the end, the recovery of nearly 45 wt% of the initial lignocellulosic biomass in form of lignin-based polymer with distinct and certainly tunable characteristics represent a promising and cost-effective methodology to recycle different lignocelullosic wastes and by-products by converting them into the technical lignin-based polymer for value-added applications.

## Data Availability

Not applicable.

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
