# Peer review of "Ambient-Pressured Acid-Catalysed Ethylene Glycol Organosolv Process: Liquefaction Structure–Activity Relationships from Model Cellulose–Lignin Mixtures to Lignocellulosic Wood Biomass"

_polymers, 2021, doi:10.3390/polym13121988_

Round 1
Reviewer 1 Report
Title: Ambient-pressured acid-catalysed ethylene glycol organosolv process: Liquefaction structure-activity relationships from model cellulose-lignin mixtures to lignocellulosic wood biomass
Authors: Edita Jasiukaityté-Grojzdek and colleagues
Several amendments should be required previously to the publication as follows:
- The descriptions on the MC cellulose and beech sawdust should be more elaborated. For example, the particle size of the sawdust is required, isn’t it? Additionally, several photographs during the sample preparation may enhance the appreciation of the readers. In the present status, it is difficult to image the experimental procedure in details. Otherwise, the photographs shown in Fig. 1 should appear immediately after Scheme 1.
- “CLR” and “WR” should not be abridged in the title of Subsection 2.2.3.
- The presentation of the weight must be unified either using “z” or “Z”.
- The definition of “X” is ambiguous from the description in the line 141. It seems to me that “X” is not CL or W but is the ratio of the weight before/after the treatment. This issue is also applicable to the definition of “YLBP”. The authors must provide the definitions precisely.
- Although Fig. 1 contains four photographs, there are no explanations on the photographs. If each photograph correspond to some item listed in Scheme 1, the correspondence should be specified.
- “Time” of the horizontal axis in Fig. 2 is too terse. The presentation must be elaborated. This issue is also applicable to Tables 1-4.
Author Response
Title: Ambient-pressured acid-catalysed ethylene glycol organosolv process: Liquefaction structure-activity relationships from model cellulose-lignin mixtures to lignocellulosic wood biomass
Authors: Edita Jasiukaityté-Grojzdek and colleagues
Several amendments should be required previously to the publication as follows:
- The descriptions on the MC cellulose and beech sawdust should be more elaborated. For example, the particle size of the sawdust is required, isn’t it?
The authors acknowledge the comments of the referee. A more detailed information regarding MC cellulose and beech sawdust has now been included in the manuscript.
- Additionally, several photographs during the sample preparation may enhance the appreciation of the readers. In the present status, it is difficult to image the experimental procedure in details. Otherwise, the photographs shown in Fig. 1 should appear immediately after Scheme 1.
The authors thank the reviewer for the suggestion. Fig. 1 has now been improved and contains more detailed information regarding each sample. Likewise, scheme 1 has been upgraded with real samples illustration to give a clearer picture of the integrated process proposed in this manuscript.
- “CLR” and “WR” should not be abridged in the title of Subsection 2.2.3.
The authors thank the referee for bringing this to their attention. This subsection title has been revised.
- The presentation of the weight must be unified either using “z” or “Z”.
The authors acknowledge the comment of the referee, nevertheless, capital “Z” has always been used. It might seem otherwise due to the equation format.
- The definition of “X” is ambiguous from the description in the line 141. It seems to me that “X” is not CL or W but is the ratio of the weight before/after the treatment. This issue is also applicable to the definition of “YLBP”. The authors must provide the definitions precisely.
The authors thank the reviewer for bringing this to our attention. Definitions have now been clarified.
- Although Fig. 1 contains four photographs, there are no explanations on the photographs. If each photograph corresponds to some item listed in Scheme 1, the correspondence should be specified.
The authors acknowledge the comments of the referee, hence Fig.1 and scheme 1 have been revised.
- “Time” of the horizontal axis in Fig. 2 is too terse. The presentation must be elaborated. This issue is also applicable to Tables 1-4.
The authors thank the referee for bringing this to their attention. Fig. 2 and Tables 1-4 have been revised.
Reviewer 2 Report
Article "Ambient-pressured Acid-catalysed Ethylene Glycol Organosolv Process: Liquefaction Structure–Activity Relationships from Model Cellulose–Lignin Mixtures to Lignocellulosic Wood Biomass" is very well described and characterized, but it is necessary to show the chemical structures of all steps, especially the reaction schemes and products described in Table 2.In addition, FTIR spectra could help to visualize and complement the list of indicated groups in this same Table. In Figure 4, the groups referring to each chemical shift in the Figure could be indicated to make it more didactic. In Figure 6 some peaks were not indicated, a better justification is needed.
Author Response
- Article "Ambient-pressured Acid-catalysed Ethylene Glycol Organosolv Process: Liquefaction Structure–Activity Relationships from Model Cellulose–Lignin Mixtures to Lignocellulosic Wood Biomass" is very well described and characterized, but it is necessary to show the chemical structures of all steps, especially the reaction schemes and products described in Table 2.
The authors acknowledge the comments of the referee. The chemical structures of all steps, the proposed reaction mechanism and end products described in this table are also represented in Figure S1 of Supporting Information.
- In addition, FTIR spectra could help to visualize and complement the list of indicated groups in this same Table.
Authors agree that FTIR spectra could help visualize and complement the described functional groups in Table 2. Nonetheless, FTIR is more a qualitative technique while NMR is an accurate and reliable method. Hence, we believe NMR quantitative and qualitative determination gave all the necessary information required to make the mentioned conclusions.
- In Figure 4, the groups referring to each chemical shift in the Figure could be indicated to make it more didactic.
The authors thank the reviewer for this suggestion. The functional groups have now been indicated in Figure 4.
- In Figure 6 some peaks were not indicated, a better justification is needed.
The authors are aware that not all peaks have been identified, which this is due to prompt reactions and the lack of all standards. Yet, all the essential standards for the sample analysis were in fact analysed and allowed all the iteractive conclusions.